# What Type of Housework Happiness Do You Prefer? Does Gender and Health Matter? A Taiwanese Study

**DOI:** 10.3390/ijerph19148409

**Published:** 2022-07-09

**Authors:** Ching-Fen Lee, Shain-May Tang

**Affiliations:** 1Minghsin University of Science and Technology, Hsinchu 30401, Taiwan; 062888@yahoo.com.tw; 2National Open University, New Taipei City 247031, Taiwan

**Keywords:** happiness, division of housework, gender, couples’ health status, gender role attitude, decision-making power

## Abstract

The purposes of this study was to discover the circumstances in which people gain happiness from performing housework and to understand gender differences in housework-related happiness. We used national data from the Taiwan Social Change Survey conducted in 2011. Only married and cohabiting respondents were included in this study (*N* = 1250). Two types of housework happiness were developed: the goal satisfaction type (GST) and the activity enjoyment type (AET), based on interview results in pilot studies and the concept of positive psychology. We found that the significant variables on the two types of housework-related happiness for the total sample were gender, socioeconomic status, gender role attitude, decision-making power, relative feminine housework, and respondent’s health. In addition, the effects on the two types of housework-related happiness for males and females are different. Most people derive happiness from housework if their preferences for type of housework and their personal characteristics are matched. It is possible to transform an otherwise monotonous daily activity into a source of happiness through the process of understanding your housework preference type, learning to enjoy the beauty of housework, and creating fun with chores for families. However, the survey (TSCS) used in this study was carried out over 10 years ago (2011) and the results may be somewhat different in Taiwan today.

## 1. Introduction

Housework is an indispensable part of family life. Unfortunately, most people regard housework as ordinary, repetitive, and boring tasks requiring little skill [1,2]. Therefore, many studies have assumed that people would prefer to avoid housework and believe that housework is a source of stress that can increase depression [3,4] and decrease well-being [5,6]. Those one-sided perspectives completely ignore the fact that chores can also be enjoyable or that some people prefer to do chores [7]. In this study, we argued that housework can bring happiness if people’s preferred circumstances are understood; in other words, if we can elucidate a source of happiness in carrying out housework, we may be able to transform the otherwise boring chores into fun family work. Therefore, two types of housework-related happiness based on positive psychology theories were operationalized in this study to investigate our perspective.

Family researchers have not focused much on the division of domestic labor in Taiwan because of a patriarchal outlook, until the 1990s. With the increase in the number of employed women and improvement in women’s education, the authority and respect commanded by a husband and wife within a family in Taiwan are no longer determined by only traditional patriarchal norms [8]. Many operational concepts and hypotheses, based on three Western perspectives (gender role, resources, and time availability), have been developed and tested in the last two decades [9]. Although differences in cultural contexts can affect the division of housework [6], there are some common findings that are similar to the results of the Taiwan study: (1) men have more traditional gender role attitudes than women; (2) men have more power and resources than women in the family and society; (3) men spend more time than women in paid work, which results in women undertaking a massive amount of housework and childcare [9].

Those three classic theories that have analyzed the inequity division of housework in the past point to an essential fundamental issue: the relationship between the family–work and gender [10,11]. Whether it is the gender role perspective that regards housework as a woman’s business [12,13], the resources perspective that emphasizes that those with resources and power can do less housework in family negotiation [14,15], or in the time availability perspective, working time often takes precedence over housework time when it comes to family schedules [16]. A new perspective of gender deviance neutralization based on Brines’ work argues that the more men rely on their wives for financial support, the fewer household chores they do [17]. Although other empirical findings do not support this new perspective, it suggests an essential effect of the relationship between economic factors and gender norms on the gendered division of housework. Furthermore, while researchers have focused on how the structure of paid work explains the gendered division of housework, others have further argued that an individual’s choice about “doing gender” often appears to be the result of social interaction [18] and job constraints [19]. Thus, the division of housework between husbands and wives is affected by complex contexts, including economic, time-availability, or gender factors, and the interactions between the three factors.

Those theories analyze housework as an unfulfilling activity and imply that people do not derive happiness from performing housework. Barnett and Shen [20] found that women’s housework is related to their psychological distress in double-income families because of the limited time available; the more time a woman spends on housework, the more she feels annoyed, with the pressure continuously increasing. Golding [21] and Glass and Fujimoto [22] also discovered that housework participation is associated with increased depression among married adults. Oshio et al. [23] investigated families in Asian countries and found that women’s marital satisfaction in China, Japan, and Korea had a consistently negative association with housework.

However, recent evidence indicates that housework may have some positive outcomes. Buettner [24] studied the sources of happiness in daily life and observed that housework, such as cooking and child care, ranked between eighth and tenth of the most-enjoyed activities. Caplan and Schooler [25] discovered that participation in household chores could increase self-confidence and reduce frustration for married women. Additionally, Tang [26] found that Taiwanese men prefer recreational housework (such as playing with kids) compared to women because such chores help alleviate their work pressures. For older adults, performing housework is indicative of independence and social participation, which are key indicators of quality of life [27]. From the perspective of mindfulness, housework can also positively affect personal mental health. For example, when washing dishes, people who take the time to smell the soap and take in the experience can reduce nervousness and it can benefit mental health [28]. Moreover, housework, such as cleaning, can help people gain a sense of control over their environment and calm their minds by engaging in the repetitive processes of housework [29].

In this study, we use the concept of positive psychology to develop the sources of happiness in carrying out housework. Positive psychology attempts to improve quality of life by utilizing such subjective experiences as individual optimism, positive emotions, and intrinsic motivation [30]. Various definitions and theories of happiness have been proposed, broadly classified into need and goal satisfaction theories, process and activity theories, and genetic and personality predisposition theories [31]. Need and goal satisfaction theories focus on the happiness attained after achieving goals and fulfilling demands. Therefore, an individual derives happiness from achieving goals, and such achievements’ outcomes are valuable and meaningful to the individual [32,33]. According to process and activity theories, people derive happiness by engaging in activities. When people engage in activities, they find them interesting and are adequately skilled to complete these activities; their sense of happiness and joy grows [34,35]. Congenital factors, which cannot be easily changed, feature highly in genetic and personality predisposition theories; these factors were not considered in this study.

Since everyone experiences happiness in different ways, such is the case in the execution of housework. Based on the concept of positive psychology, some people may tend to feel happy by carrying out housework to meet their needs and goals. In contrast, others feel happy in the process and activities of carrying out housework. This study aims to understand what kind of happiness we obtain from carrying out housework. In the future, we can strengthen the characteristics of this happiness in carrying out housework, thereby improving the well-being of housework.

In addition to considering several variables emphasized by traditional housework theories (e.g., gender role attitudes, working hours, power, and socioeconomic status), health status and gender are the items we focused on. Although past research has demonstrated that personal health status [5,36,37] and a spouse’s health status [38] have central roles in improving personal happiness and quality of life, health issues are rarely considered in studies of relations between housework participation and personal well-being. Thus, in the present study, we explored gender differences in “housework-related happiness” and proposed that individual health and partner health status can influence housework happiness.

Many studies have found gender to be the critical factor in the differences in housework participation and effect [39,40]. Past studies have found significant differences between the genders in the amount of time they spend on housework [41], the chore items they participate in [42], and their attitude towards housework [43]. In addition, the relative share of housework negatively correlates with women’s, but not men’s, marital satisfaction in Taiwan [6]. More recently, a critical view on the division of housework has argued that neither resource nor time availability perspectives are gender-neutral and can be explained using gender perspectives [7]. A study across east Asian countries has also found gender differences in the determinants of happiness [44]. Therefore, this study will analyze the factors that may affect the type of housework happiness in different genders.

## 2. Methods

### 2.1. Sample Selection and Ethical Considerations

In this study, we used data from the 2011 Taiwan Social Change Survey (TSCS). The TSCS is a cross-sectional social survey that has been conducted every 5 years since 1984. The data archive of TSCS is an open-access database for academic use only. To access the database, applicants submit a statement describing their motivation and obtain approval from the Survey Research Data Archive at Academia Sinica (the national academy of Taiwan), which administers the database. A multistage stratified random sampling method was used to generate a nationally representative adult population sample. The sampling process was based on a stratified three-stage probability proportional-to-size approach to ensure that each person in the population had an equal chance of being selected [45]. The target population of this study was Taiwanese citizens aged 18 or older with registered domiciles in Taiwan; the number of interviewees totaled 2135. In 2011, the topical module of the second wave of the sixth phase of TSCS was on family. It was ideal for our analysis since we have participated in the question design of this data and added the question about sources of housework happiness to this survey. It also included information on respondent characteristics, such as age, socioeconomic status (counted by education and occupation), working hours, self-reported health, and spouses’ health status, as well as their involvement in decision-making power and gender role attitude. Only married and cohabiting respondents were included in this study, thus reducing the study sample to 1250.

### 2.2. Outcome Variable-Type of Housework-Related Happiness

Before designing this questionnaire, we asked 20 married men and women in a pilot study about their views on the sources of happiness in carrying out housework. The question was designed based on the interview results and the positive psychology concept. The housework-related happiness was operationalized using the question: “In which of the following situations do you feel the happiest when performing housework?” Respondents selected one of the following answers: “When receiving recognition, gratefulness, or compliments from family members,” “When I feel comfortable after completing household chores,” “When I can get together with my family,” “When I help decrease my spouse’s burden,” “When enjoying doing the housework itself,” “When considering housework as a leisure activity,” “Never feel happy doing housework,” and “other.”

According to goal satisfaction theories, happiness is gained when a critical goal is achieved. Therefore, four items: “When receiving recognition, gratefulness, or compliments from family members,” “When I feel comfortable after completing household chores,” “When I can get together with my family,” and “When I help decrease my spouse’s burden,” defined as the goal satisfaction type (GST). By contrast, activity theories state that happiness is derived when an individual engages in preferred activities. Thus, “When enjoying doing the housework itself” and “When considering housework as a leisure activity” comprised the activity enjoyment type (AET). We also excluded respondents who chose “Never feel happy doing housework” and “other” in this study.

### 2.3. Independent Variables

Age was measured using the birth year that was deducted from the survey year (2011) to calculate respondents’ ages.

Socioeconomic status (SES) was operationalized using the two-factor SES index modified from Hollingshead and Redlich [46] by Lin [47]. The index is calculated as the sum of two scores: (education level × 4) + occupation level × 7). The higher the sum, the higher the personal SES. Education (What is your education level?) was ranked into six levels, from 1 (elementary school education, self-study, no education, or illiterate) to 6 (a master’s or doctoral degree). Occupation (What is your main occupation (at present, before retirement, or before exiting the labor market)?) was ranked into five levels, as recommended by Lee and Hwang [48]: (a) 1—farmers, forest workers, fisherfolk, animal husbandry workers, and unskilled and manual laborers; (b) 2—service and sales professionals and skilled workers; (c) 3—transactional support staff; (d) 4—technicians and associate-level professionals; and (e) 5—senior management staff and professionals.

Working hours were measured using the following question: “On average, how many hours per week do you work at your job?” Higher scores indicated that the respondents had longer working hours.

Gender role attitude was operationalized using responses to the question, “During an economic recession, is it appropriate for women to be laid-off before men?” The respondents responded on a 5-point scale ranging from 1 (strongly disagree) to 5 (strongly agree). Therefore, higher scores were indicative of a more traditional attitude toward gender roles.

*Decision-making power* was measured using the following question: “Who can decide the purchase of higher-cost items at home between the couple?” Responses were rated on a scale of 1 (always my spouse) to 5 (always myself); thus, higher scores indicate that the respondents have more power in familial decision-making than their spouses.

Division of housework was gauged using responses to the question, “During the past year, how often did you and your spouse do the following?”: prepare dinner, laundry, cleaning, and small repairs. The respondents rated each of the four chores on a 7-point scale ranging from 1 (never) to 7 (almost every day). The relative housework score was operationalized using respondents’ scores minus their partners’ scores; therefore, for a couple, a relatively higher score indicates that the respondent performs more housework than his/her partner. The first three listed chores are typically performed by women and the fourth chore by men [40,43,49]. These two gender-based types of domestic labor vary in their features and societal expectations regarding the frequency, flexibility, necessity, and substitutability of work [43]. Therefore, these variables represented relative feminine and relative masculine housework in this study.

Respondent’s and partner’s health were measured using the following questions: “How would you rate your health?” and “How would you rate your partner’s health?” Responses were rated on a scale of 1 (very poor) to 5 (very good), with higher scores indicating that the respondents and respondents’ partners were in better health.

## 3. Results

### 3.1. Characteristics of the Participants

About 53.5% (*n* = 669) of the respondents were men and 46.5% (*n* = 581) were women. The respondents ranged from 21 to 91 years old, with a mean age of 54.07 years (SD = 13.74). Additionally, 27.7% (*n* = 346) of the respondents had received only elementary school education, had engaged in self-study, or were illiterate; 13.8% *(n* = 173) had received a junior high school education; 27.5% (*n* = 344) had received a high school education; 25.8% (*n* = 323) had graduated from a junior college or university; and 5.1% (*n* = 64) had received a Master or Doctor degree. The respondents’ average working hours per week was 48.68 (SD = 15.28) hours. On average, most respondents tended to disagree with traditional gender role attitudes and have equal decision-making power between spouses. The mean gender role attitude was 2.13 (SD = 1.00; range: 1–5). The mean decision-making power was 3.03 (SD = 1.19; range: 1-5). Moreover, for housework, the mean of respondent’s feminine housework was 14.10 (SD = 5.60; range: 3–21), and their partner’s feminine housework was 13.73 (SD = 6.24; range: 3–21). The mean of respondents’ masculine housework was 2.73 (SD = 1.38; range: 1–7), and their partner’s masculine housework was 2.42 (SD = 1.35; range: 1–7). Almost three-fourths of the respondents rated their health as good (54.8%) or very good (18.1%), and 15.9% reported having poor health (14.6% for poor; 1.3% for very poor). In addition, three-quarters of respondents indicated their partner’s health as good (53.5%) or very good (20.4%), and 16.6% reported their partner having poor health (14.9% for poor; 1.7% for very poor). The descriptive characteristics of participants are presented in Table 1.

### 3.2. Source of Housework Happiness

Among the activity enjoyment type (AET), respondents felt happy when they considered housework to be a leisure activity (51.7%, *n* = 646), or when they enjoyed carrying out the housework itself (5.0%, *n* = 63). Among the goal satisfaction type (GST), the respondents felt happy when their housework performance helped decrease their spouse’s burden (12.6%, *n* = 158), when they were comfortable after completing household chores (10.2%, *n* = 128), when they were able to get together with their family (9.9%, *n* = 124), or when they received recognition, appreciation, and praise from their family (5.3%, *n* = 66). Three percent *(n* = 37) of the respondents indicated that they never felt happy when carrying out housework, and were excluded from further analyses. Therefore, we found that 56.7% (*n* = 709) and 38.1% (*n* = 476) of the respondents derived housework happiness when they were engaging in AET and GST; the remaining 2.2% (*n* = 28) did not indicate their choices and were considered as a missing value.

### 3.3. Results of Binary Logistic Regression

The results of the multivariate logistic regression are presented in Table 2. The independent variables with a significant effect on the two types of housework-related happiness (AET and GST) for the total sample (model 1) were gender (OR 0.53; 95% CI 0.33–0.86), SES (OR 1.01; 95% CI 1.00–1.02), gender role attitude (OR 1.27; 95% CI 1.11–1.46), decision-making power (OR 1.18; 95% CI 1.06–1.32), traditionally feminine housework (OR 1.03; 95% CI 1.01–1.05) and respondent’s health (OR 0.83; 95% CI 0.71–0.98). The results of the total sample model specification revealed that female respondents were more likely to prefer AET housework and male respondents were more likely to prefer GST housework than were their opposite-gender counterparts. Notably, the respondents who preferred AET housework tended to have a significantly higher SES, a more traditional gender role, a poorer health, and more household decision-making power than their spouse, compared with the respondents who preferred GST housework, these respondents also generally performed traditionally feminine housework.

According to model 2, the independent variables with a significant effect on the two types of housework-related happiness (AET and GST) for the male responses were SES (OR 1.02; 95% CI 1.00–1.03), gender role attitude (OR 1.29; 95% CI 1.07–1.56), decision-making power (OR 1.17; 95% CI 1.01–1.36), and relative masculine housework (OR 1.13; 95% CI 1.01–1.26). The male respondents who preferred AET housework tended to have a significantly higher SES, a more traditional gender role, and more household decision-making power than their spouse, compared with the respondents who preferred GST housework; these respondents also generally performed relative masculine housework.

According to model 3, gender role attitude (OR 1.24; 95% CI 1.00–1.52) and feminine housework (OR 1.05; 95% CI 1.01–1.08) significantly affected female respondents’ housework-related happiness, while respondent’s health (OR 0.78; 95% CI 0.60–1.00) was very closed to the critical point of significance. The female respondents who preferred AET housework tended to have a more traditional gender role and perform more relative feminine housework. If the respondent’s health was considered, we could find that the female respondent who preferred GST tended to have better health.

## 4. Discussion

### 4.1. Housework Can Be a Happy Activity

Only 3% of our respondents reported never deriving happiness from household chores. Although this does not indicate that the other 97% of respondents enjoy housework, it does suggest that a vast majority of them can experience happiness in performing housework under certain circumstances. This study does not overlook the widely held view that people are annoyed with and feel antipathy toward housework. Rather, we explored housework-related happiness from different angles and considered people’s daily housework routines under a new and positive definition. Given our results, researchers may need to reexamine previous studies that have regarded housework as a solely negative task.

Nearly 60% of the respondents were determined to experience happiness when engaging in AET housework. In other words, these respondents feel happier about performing housework if they can find ways to transform that housework into something engaging, an idea which reflects Csikszentmihalyi’s [34] flow theory. Similarly, Robinson and Milkie [50] noted that many people do not loathe housework and that cooking, in particular, is a stimulating and fun housework activity, and Tang [9] asserted that employed women occasionally regard housework (especially those activities relating to cleaning or house decoration) as a leisure activity because it can alleviate work pressures. Hanley et al. [28] also found that if people could take the time to smell the soap and to take in the experience of washing dishes, performing housework can reduce nervousness and benefit mental health. The other 40% of the respondents were classified as achieving happiness when engaging in GST housework. Need and goal satisfaction theories, which emphasize that happiness is gained from the achievement of goals and fulfillment of demands, apply to this group [32,33]. Therefore, for this group of respondents, understanding the personal value and meaning of housework for their families and themselves is critical.

In general, we also found that the respondents who preferred AET housework tended to be female, have a higher SES, prefer traditional gender roles, have more decision-making power, and have poorer health, compared with the respondents who preferred GST housework. In addition, those who preferred AET housework were more likely to perform relative feminine housework than were those who preferred GST housework. These results show that people can derive happiness from housework if their personal characteristics and types of housework happiness can be matched.

Because of the traditional values that men belong to work roles and women belong to family roles still exist in Taiwan, women still undertake most of the housework regardless of whether they are employed or not [9]. In Taiwan, women perform 3.4 times more housework than men [6]. When the findings echo the Taiwanese social context, it is understandable that when women’s responsibilities involve completing the majority of housework, their happiness needs to come from the enjoyment and leisure of the moment and process of carrying out housework.

These findings should be applied to design programs that improve people’s well-being in household chores. For example, some chores can be designed as leisure activities for family members to increase the interest of housework; family members can learn to express gratitude more often, so that those who carry out housework can have positive feedback; or by changing our attitude towards housework through courses to enjoy the moment of performing housework, as well as have a sense of accomplishment when complete.

### 4.2. Gender Differences in Deriving Happiness from Housework

The factors influencing housework-related happiness were found to differ between men and women, with only gender role attitude affecting both the male and female respondents. Specifically, the male respondents’ housework-related happiness was influenced by their SES, gender role attitude, decision-making power, and the amount of relative masculine housework performed. In contrast, the female respondents were influenced by the amount of relative feminine housework performed and gender role attitudes.

As previous research has revealed, men with high SES, decision-making power, and a traditional gender role attitude, tend to be authoritative, and rarely perform housework because they can use their power in exchange for partner’s housework execution [51], unless they are interested in housework participation [26].

In addition, men with traditional values regard housework as a women’s duty; on the other hand, liberal men, who are more willing to share housework [52], may highly value the familial interaction that housework facilitates. At the same time, their happiness increases when they engage in more housework, because their participation in housework can reduce their guilt that their spouses still carries out a lot of housework in Taiwan [6]. Hence, liberal men derive happiness from housework participation as it helps them achieve and satisfy personal and familial goals and needs.

In Taiwan, the cost of house maintenance or repair is not high, and the families do not teach children to learn some simple maintenance work. A repair man will usually be hired to perform the job. Thus, men who are willing to carry out such work are primarily out of for personal preference or personal interest [43]. Therefore, male respondents who engaged in more masculine housework tended to have more AET tendencies than those who carried out less such work.

Although Taiwanese men have more traditional gender role attitudes than women, the values of gender roles in Taiwanese society are still conservative and traditional [53]. Female respondents who do more feminine chores and have more traditional gender values tended to be more AET tendencies than were those who did less such work and had less traditional gender value. It is not surprising that women, who are still bound by traditional gendered values in Taiwan derive happiness from housework because they can perform their chores according to their preferences rather than achieving social or familial expectations.

### 4.3. Influence of Health on Housework Happiness

In general, we also found that the respondents who preferred AET housework tended to have poorer health than those who preferred GST housework, since people with poor health can use it as an acceptable reason for not carrying out housework; unless people are interested in it or enjoy the pleasure it brings. The female respondents’ health was slightly related to the types of housework happiness. Specifically, healthy women tended to gain happiness from GST housework, whereas unhealthy women tended to gain happiness from AET housework. Housework is still primarily performed by women [54], and a person’s health is relevant to their ability to perform housework [55]. Therefore, a woman who is unhealthy but must complete household duties may be more likely to feel happy when performing housework if the housework is viewed as a personal interest rather than a responsibility. By comparison, a healthy woman is less stressed when conducting housework. She, therefore, may be more likely to feel happy when performing housework if she can fulfill the housekeeping role expected by society and make life easier for herself and her family.

### 4.4. Limitations & Future Directions

Although this study proved that housework could be associated with positive well-being, if people’s preferred circumstances were understood, the circumstances people prefer may change with the development of family life stages and the differences in cultural contexts. Unfortunately, the data we used in this study did not have such information, and it is suitable for consideration in future studies. The other limitation was the survey (TSCS) used in this study was carried out over 10 years ago (2011), and the results may be somewhat different in Taiwan today. Meanwhile, from the perspective of several housework studies, the lack of possible positive effects of housework performance is a direction for future research.

Gender still plays an essential role in this study, and as Dominguez-Folgueras [7] mentioned, many variables related to housework are ultimately related to gender. This study not only found the influence of gender differences (women prefer AET, men prefer GST), but also found that gender role attitudes, power, and housework that affected the type of housework-related happiness were closely related to gender. Therefore, future research on housework still needs more attention on the effect of gender.

Working hours were not significantly related to housework-related happiness in this study. Work conditions, such as a full-time job or not, night shift or day shift, work flexibility, off-duty time, and job stress, could be considered for future research because work conditions are associated with housework division and marital satisfaction [11,56]. In addition, we did find that relative feminine and masculine housework were associated with housework-related happiness. However, the time devoted to housework can also affect marital relationships [57], which may affect housework-related happiness so that absolute housework time can be considered for future research.

The contemporary trend of aging societies spans the globe, and health will undoubtedly be an essential factor for investigating housework and happiness among elderly families in the future [58], as well as continue to be a key factor influencing their quality of life. Notably, previous studies on housework division have rarely mentioned how a couple’s health is influenced. In a family system, an individual’s and partner’s health should be considered when examining housework-related happiness. Finally, this study provides a positive psychological perspective on housework, but how families learn, reflect, and apply this perspective to the family system still needs a broad push from practitioners, such as family life educators.

## 5. Conclusions

The results showed that housework can be a happy activity, since only 3% of our respondents reported never deriving happiness from household chores. And, nearly 60% of the respondents who preferred AET housework tended to be female, have a higher SES, prefer traditional gender roles, have more decision-making power, and have poorer health, compared with the respondents who preferred GST housework. We also found that both gender and health affected the housework-related happiness. The male respondents’ housework-related happiness was influenced by their SES, gender role attitude, decision-making power, and the amount of relative masculine housework performed. In contrast, the female respondents were influenced by the amount of relative feminine housework performed and gender role attitudes. Meanwhile, the respondents who preferred AET housework tended to have poorer health than those who preferred GST housework. Therefore, understanding the various implications of housework performance with using the positive perspective can be a direction for future research. Also, research on housework needs to pay more attention to gender and health effects.

## Figures and Tables

**Table 1 ijerph-19-08409-t001:** Characteristics of the sample.

Variables	*n*	(%)	Mean	(SD)
Gender				
Male	669	(53.5%)		
Female	581	(46.5%)		
Age			54.07	(13.74)
Education level				
None/Illiterate/Elementary school	346	(27.7%)		
Junior high school	173	(13.8%)		
High school	344	(27.5%)		
College/University	323	(25.8%)		
Master/Doctor	64	(5.1%)		
Working hours			48.68	(15.28)
Gender role attitude			2.13	(1.00)
Decision-making power			3.03	(1.19)
Respondent feminine housework			14.10	(5.60)
Partner feminine housework			13.73	(6.24)
Respondent masculine housework			2.73	(1.38)
Partner masculine housework			2.42	(1.35)
Respondent’s health				
Very poor	16	(1.3%)		
Poor	182	(14.6%)		
Neither good nor poor	141	(11.3%)		
Good	685	(54.8%)		
Very good	226	(18.1%)		
Partner’s health				
Very poor	21	(1.7%)		
Poor	186	(14.9%)		
Neither good nor poor	117	(9.7%)		
Good	669	(53.5%)		
Very good	255	(20.4%)		

**Table 2 ijerph-19-08409-t002:** Binary logistic regression predicting two types of housework-related happiness.

Meanings	Model 1 (Total)			Model 2 (Male)			Model 3 (Female)		
	Odds Ratio	*p*-Value	95%CI	Odds Ratio	*p*-Value	95%CI	Odds Ratio	*p*-Value	95%CI
Gender(0 = female)	1.00		1.00						
Male	**0.53**	0.011	0.33, 0.86						
Age	0.99	0.172	0.98, 1.00	0.99	0.121	0.99, 0.97	1.00	0.675	0.98, 1.02
SES	**1.01**	0.021	1.00, 1.02	**1.02**	0.012	1.00, 1.03	1.01	0.592	0.99, 1.02
Working hours	1.00	0.528	1.00, 1.01	1.00	0.652	0.99, 1.01	1.01	0.087	1.00, 1.02
Gender role attitude	**1.27**	0.001	1.11, 1.46	**1.29**	0.009	1.07, 1.56	**1.24**	0.048	1.00, 1.52
Decision-making power	**1.18**	0.004	1.06, 1.32	**1.17**	0.031	1.01, 1.36	1.19	0.057	1.00, 1.42
Relative feminine housework	**1.03**	0.004	1.01, 1.05	1.02	0.134	1.00, 1.05	**1.04**	0.013	1.01, 1.08
Relative masculine housework	1.02	0.630	0.94, 1.11	**1.13**	0.039	1.01, 1.26	0.91	0.120	0.80, 1.03
Respondent’s health	**0.83**	0.025	0.71, 0.98	0.88	0.219	0.72, 10.8	0.78	0.051	0.60, 1.00
Partner’s health	1.06	0.456	0.91, 1.23	0.95	0.602	0.77,1.16	1.19	0.136	0.95, 1.50
Constant	1.22	0.744		0.79	0.778		0.76	0.771	
Omnibus χ2	**90.20** (*p* = 0.000)			**31.20** (*p* = 0.000)			**20.98** (*p* = 0.013)		
Hosmer & Lemeshowχ2	10.32 (*p* = 0.24)			8.92 (*p* = 0.349)			2.28 (*p* = 0.971)		
N	1083			577			506		

Values in bold indicate significant relationships (*p* < 0.05).

## Data Availability

Ying-hwa Chang (2015). 2011 Taiwan Social Change Survey (Round 6, Year 2): Family (C00222_1) [data file]. Available from Survey Research Data Archive, Academia Sinica. https://doi.org/10.6141/TW-SRDA-C00222_1-1.

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
