# Peer review of "What Type of Housework Happiness Do You Prefer? Does Gender and Health Matter? A Taiwanese Study"

_ijerph, 2022, doi:10.3390/ijerph19148409_

Round 1

Reviewer 1 Report

The topics of the article are interesting and stimulating. The article is well written. At the same time, there are some problems. The first one is related to the description of the frame (time frame, country frame, labor market frame, gender equality frame, etc). The issue of reconciliation between family and work is relevant to this topic. At the same time, these things are fundamental to understanding and explaining the meaning of happiness in relation to housework. 

Moreover, the data are old, and it's necessary to justify this choice. At the same time, a country's background it's useful to understand the results and the analysis.

Reviewer 2 Report

This paper presents a new approach to housework from a gender perspective: happiness. This psychological perspective enlights the research on house chores and gender division within the family because it considers housework as a positive activity, which can make people happy. However, there is a mistake from the beginning. Women are unhappy with housework because all these tasks are compulsory for them, and they have to do it or nobody does it at home. In the case, this women have children or other dependent adults, they have to face the ethical dilemma: helping their relatives that have real needs, or focusing on their work outside the home.

The way to correct this mistake would be to ask people if after a long working day when men and women are back at home, do they have to keep working or do they can rest? So, authors should take into account that variable too. How many hours do people work outside the home, and at what time they are back home to start their housework? These variables are key factors to understand the two types of housework happiness, GST and AET. When you are back home and you are tired because you worked during the whole day, happiness could mean resting. Then, there is no chance for a positive perspective about homework.

There is a lot of literature about the double shift and the third shift of working mothers, this phenomenon affects women and it is an obstacle to developing their professional careers.

Finally, I would say there is a little spelling mistake on page 6. It should say “to prefer AET housework” instead of “to prefer AEY housework”.

Reviewer 3 Report

Der Authors,

I am I happy that I had the opportunity to read an interesting article. I have, however, several notes:

The research paper’s stated purpose is to discover the circumstances in which people gain happiness from performing housework and to understand gender differences in house- work-related happiness. The purpose sounds to be clear and meaningful.

The literature review covers the essential areas of focus of the paper. However, I recommend adding a section on different approaches to the perception of the division of domestic work between men and women. You started line 40 with words: Because of this patriarchal outlook .... While it is implicitly clear that you are referring to the unequal division of labour between men and women in households, the literature review itself does not imply this.

For the article, you use data from the 2011 Taiwan Social Change Survey. As you stated, survey is conducted every 5 years. This means that two more surveys have already been carried out since 2011. What is the reason for analysing data from 2011 when more recent data is already available? What makes the 2011 survey collection special?  Was the module on family only included in the 2011 survey?

The relevance of Section 2.2 Outcome Variables .... to the overall description of the methodology is not clear. It is also unclear whether you excluded respondents who answered "Never feel happy doing housework" from the total study sample (1250) or from the pilot study respondents.

Results concerning model 2 (first sentence, lines 240 – 242) are not complete.

I highly recommend to clearly identify dependent variables in regression models, as it is unclear how two types of housework-related happiness (AET and GST) are coded in the models (hence the interpretation of the models is also ambiguous).

In terms of the overall focus of the article, I recommend framing the discussion more with links to Taiwan's traditions and to potential changes in perceptions of happiness in the context of housework.

Limitation of the research as well as the implications of the findings are absent (unclear) in the paper.

Minor formal comments:

-          there are two subsections 2.2 (line 131 and line 148) in the text, using a different font

-          there are several formal textual flaws in the article (e.g. missing space between a word and a parenthesis)

-          SES is not defined in the article (although it is clear that it is socioeconomic status, I recommend defining the abbreviation in the article)

Reviewer 4 Report

The paper "What type of housework happiness you prefer? Does gender and health matter?" studies the circumstances in which people gain happiness from performing housework with a focus on gender differences.

In general, the paper is clear and the different sections are well discussed. However, I suggest making some changes that would improve the paper:

-          - An extended explanation of the two types of housework happiness: the goal satisfaction type (GST) and the activity enjoyment type (AET). Why these two types?

-         -  The amount of time devoted to domestic work will probably have an impact on the satisfaction with this work.

-          - An extended discussion of results including a comparison of studies in other countries.

Round 2

Reviewer 2 Report

The authors have introduced different changes according to the previous reviews. So, I would say the paper could be published right now.

Author Response

We greatly appreciated your recommendation for our research publication.